# Butyrate in Human Milk: Associations with Milk Microbiota, Milk Intake Volume, and Infant Growth

**DOI:** 10.3390/nu15040916

**Published:** 2023-02-11

**Authors:** Laurentya Olga, Janna A. van Diepen, Maciej Chichlowski, Clive J. Petry, Jacques Vervoort, David B. Dunger, Guus A. M. Kortman, Gabriele Gross, Ken K. Ong

**Affiliations:** 1Department of Paediatrics, University of Cambridge, Cambridge CB2 0QQ, UK; 2Medical and Scientific Affairs, Reckitt/Mead Johnson Nutrition Institute, Evansville, IN 47721, USA; 3Department of Agrotechnology and Food Sciences, Wageningen University, 6708 WE Wageningen, The Netherlands; 4MRC Epidemiology Unit, Wellcome Trust-MRC Institute of Metabolic Science, NIHR Cambridge Comprehensive Biomedical Research Centre, Cambridge Biomedical Campus, University of Cambridge, Cambridge CB2 0SL, UK; 5NIZO Food Research BV, 6718 ZB Ede, The Netherlands; 6Institute of Metabolic Science, University of Cambridge, Cambridge CB2 0QQ, UK

**Keywords:** human milk butyrate, human milk microbiota, human milk intake volume, butyrate intake, infant growth, infant adiposity, infant weight gain

## Abstract

Butyrate in human milk (HM) has been suggested to reduce excessive weight and adipo-sity gains during infancy. However, HM butyrate’s origins, determinants, and its influencing mechanism on weight gain are not completely understood. These were studied in the prospective longitudinal Cambridge Baby Growth and Breastfeeding Study (CBGS-BF), in which infants (*n* = 59) were exclusively breastfed for at least 6 weeks. Infant growth (birth, 2 weeks, 6 weeks, 3 months, 6 months, and 12 months) and HM butyrate concentrations (2 weeks, 6 weeks, 3 months, and 6 months) were measured. At age 6 weeks, HM intake volume was measured by deuterium-labelled water technique and HM microbiota by 16S sequencing. Cross-sectionally at 6 weeks, HM butyrate was associated with HM microbiota composition (*p* = 0.036) although no association with the abundance of typical butyrate producers was detected. In longitudinal analyses across all time points, HM butyrate concentrations were overall negatively associated with infant weight and adiposity, and associations were stronger at younger infant ages. HM butyrate concentration was also inversely correlated with HM intake volume, supporting a possible mechanism whereby butyrate might reduce infant growth via appetite regulation and modulation of HM intake.

## 1. Introduction

Butyrate is a short-chain fatty acid (SCFA) detectable in human milk (HM) [1], with concentrations ranging from 0.1–0.75 mg/100 mL between studies [1,2,3]. This four-carbon fatty acid is reported to have anti-inflammatory properties and may be protective against obesity and insulin resistance [4]. Animal studies in mice and rats showed that butyrate could improve insulin sensitivity and metabolic dysfunctions caused by exposure to a high-fat diet [5,6,7].

Butyrate is synthesized in the gut by anaerobic bacteria through the fermentation of nondigestible carbohydrates. Compared to other SCFAs, such as propionate and acetate, butyrate is reported as the greatest source of energy used by colonic epithelial cells [8]. In infants, potential origins of butyrate are either oral intake, e.g., through HM [1] or solid food, or production by bacterial fermentation of dietary compounds in the colon, presumably human milk oligosaccharides (HMOs) in infants receiving HM [9]. In contrast to intestinal butyrate, the origin of HM butyrate is not well elucidated. One possibility is that maternal gut microbiota produces butyrate, which might then enter HM via maternal circulation (since butyrate could modulate gut barrier function and affect systemic immune response) [3]. There is also evidence that HM butyrate might be produced locally by in situ HM microbiota [10]. There is yet limited evidence on the maternal and pregnancy-related factors that might influence HM butyrate concentration.

Regarding the potential benefit of HM butyrate for infants, we previously reported that HM SCFA concentrations, including butyrate, in a study cohort of over 600 lactating mothers were negatively associated with infant adiposity gains [1]. This, with several other studies, concluded that HM butyrate might prevent excessive weight and adiposity gains and reduce later obesity risk [1,11,12]. Consequently, HM butyrate intake might enable beneficial metabolic outcomes via slower growth and adiposity gains among infants receiving HM vs formula [11,13].

However, HM butyrate concentration does not necessarily represent total actual intake by infants. Measurement of absolute butyrate intake via HM is important to understanding mechanistic links with weight and adiposity gains. This study aimed to (1) provide evidence on the origin of HM butyrate by examining its associations with HM microbiota composition, (2) explore the maternal and antenatal factors associated with HM butyrate concentration, and (3) examine the associations between HM butyrate concentration, HM butyrate intake, HM intake volume, and infant growth and adiposity.

## 2. Materials and Methods

### 2.1. Study Design and Population

The Cambridge Baby Growth and Breastfeeding Study (CBGS-BF, 2015–2019) was a longitudinal prospective cohort aiming to identify factors in HM that may influence the rate of infant growth and hence alter obesity risk later in life. Parameters of HM intake and composition were measured, including HM intake volume using a deuterium-labelled water technique; repeated longitudinal HM collection and composition analyses, including macronutrients, butyrate, and HMOs; and explorative analyses of microbiota in HM and infant guts.

The study design has been reported previously [14] (Appendix A). In brief, recruitment of mother–infant pairs took place at birth at the Rosie Maternity Hospital, Cambridge, England. Strict inclusion/exclusion criteria were applied: to mothers, including intention to breastfeed from birth until at least 6 weeks of age, nonobese body mass index (HMI) [15] before pregnancy (<30 kg/m^2^), no significant illness before/during pregnancy, no antibiotic or steroid consumption in the 30 days prior to delivery, and no regular consumption of probiotics; and to infants, including singletons, born at term via vaginal delivery, with birthweight >–1.5 sex- and gestational age-adjusted SDS according to the UK 1990 growth reference [16,17].

Research clinic visits were conducted at birth, 2 and 6 weeks, and then 3, 6, and 12 months, mostly at the research facility at the hospital, or at home if not feasible. Each infant clinic visit was scheduled based on the exact age of infants with +8 days tolerance for birth, 2-, and 6-weeks visits, and +28 days for 3 months onwards.

The study was approved by the National Research Ethics Service Cambridgeshire 2 Research Ethics Committee (IRAS No 67546, REC No 11/EE/0068, original date of ethical approval 31 March 2011, date of amendment approval 7 July 2015). All mothers provided informed written consent for themselves and their infants.

### 2.2. Anthropometry

Birth weight was recorded from the medical records postdelivery. At all other time points, infants were weighed naked without nappies and before feeding using a Seca 757 electronic baby scale (Seca Ltd., Hamburg, Germany) to the nearest 1 g. Infant supine length was measured using a Seca 416 infantometer (Seca Ltd., Hamburg, Germany) to the nearest 0.1 cm.

To assess subcutaneous fat at various regions and to estimate relative subcutaneous body fat [18], skinfold thickness (SFT) was measured at 4 sites (triceps, subscapular, flank, and quadriceps) in triplicate on the left-hand side of the body using a Holtain Tanner/Whitehouse Skinfold Caliper (Holtain Ltd., Crymych, UK).

All anthropometry and body composition measurements were performed by one of three trained paediatric research nurses.

### 2.3. HM Sample Collection

For butyrate analysis, self-collected postfeed HM samples (usually 10–15 mL) were provided by mothers using hand or electric breast pumps at each visit, from birth/colostrum until 12 months if mothers were still breastfeeding, either exclusively or partially. All samples were kept frozen at −20 °C until the time of analysis.

To study HM microbiota composition, a complete HM expression from one breast was collected at 6 weeks infant age using a breast pump. Mothers cleaned the breast using antiseptic liquid, dried it with sterile paper towels, and discarded a few drops of HM prior to sample collection.

### 2.4. HM Butyrate Analysis

HM samples were defrosted and thoroughly homogenised before assays. The homogenate (400 μL) was mixed with CDCl3 solvent (400 μL) for 10 min prior to centrifugation (30 min, 10,000 rpm). The resulting nonpolar fraction was used to measure SCFAs using ^1^H-Nuclear magnetic resonance (NMR) spectra. Butyrate quantification was conducted as described previously [1].

### 2.5. HM Intake Volume

The volume of HM received by the infant was estimated using the dose-to-the-mother deuterium-oxide (^2^H_2_O) turnover technique [19]. When infants were approximately 4 weeks of age, mothers were given deuterium-enriched (tracer) water to drink, which would be incorporated into HM and passed to the infant during breastfeeding. Urine samples were collected from both mothers and infants daily for a period of 2 weeks. ^2^H enrichment in the urine samples was measured by isotope ratio mass spectrometry as described previously [19].

### 2.6. HM Microbiome Analysis

#### 2.6.1. DNA Extraction from HM Samples

HM samples were thawed at room temperature (RT), and 0.5 mL milk was added to a 2.0 mL screw cap tube containing 0.5 g of sterilised 0.1 mm zirconia beads and 0.5 mL lysis buffer (500 mM NaCl, 50 mM Tris-HCl (pH 8.0), 50 mM EDTA, 4% SDS). After mixing, 500 μL phenol and 200 μL chloroform were added. The suspension was thoroughly mixed and the FastPrep instrument (MP Biomedicals, Santa Ana, CA, USA) was used for lysis at 5 m/s for 2 times 40 sec at room temperature, with in between cooling on ice for 1 min. Thereafter, samples were centrifuged at 16,000× *g* for 5 min at 4 °C. The resulting water phase was transferred to a fresh tube; 250 μL phenol and 250 μL chloroform were added and thoroughly mixed, and samples were centrifuged (16,000× *g*, 5 min, 4 °C); this process was then repeated twice. The resulting water phase was again transferred to a fresh tube, mixed with 250 μL of chloroform, and centrifuged at 16,000× *g* for 5 min at 4 °C. Next, the final water phase (+/− 500 μL) was transferred to a fresh tube and added with 2 μL of 10 mg/mL RNase A (Qiagen, diluted in TE buffer), and the mixture was incubated at 37 °C for 15 min. Subsequently, the DNA was purified (mag mini kit, LGC Biosearch Technologies, Middlesex, UK) according to the following protocol: 400 μL of the RNase-treated final water phase was transferred to 1.5 mL tubes containing 800 μL binding buffer and 10 μL magnetic beads and mixed by pipetting. The mixture was shaken (30 min, 700 rpm, RT), and the supernatant was removed using magnetic separation (1 min). The magnetic beads were washed with 200 μL Wash Buffer 1 using gentle mixing and incubated at RT for 5 min, and supernatant was removed by magnet separation; the washing procedure was repeated with Wash Buffer 2. The magnetic beads were shaken and dried (10 min, 500 rpm, 55 °C). Next, 63 μL elution buffer was added, and tubes were incubated at 55 °C for 15 min, whilst being shaken at 9500 rpm. Then, tubes were placed in the magnet separator for 3 min, and 50 μL of the elution buffer was transferred to a fresh tube. Finally, DNA was stored at −20 °C until further processing.

#### 2.6.2. Nested-PCR Amplification of 16S rRNA Gene from HM DNA Samples

Using a 3-step nested-PCR, barcoded amplicons from the V3–V4 region of 16S rRNA genes were generated (see library PCR below for description of the third PCR step). For initial amplification of the V3–V4 part of the 16S rRNA, universal primers were used with the following sequences: forward primer, ‘5-ACTCCTACGGGAGGCAGCAG’ (broadly conserved bacterial primer 338F) and reverse primer, ‘5- CRRCACGAGCTGACGAC’ (broadly conserved bacterial primer 1061R). The PCR amplification mixture contained: 2 μL breast milk sample DNA, 0.1 μL forward primer (10 μM), 14 μL master mix (1 μL KOD Hot Start DNA Polymerase (1 U/μL; Novagen, Madison, WI, USA), 5 μL KOD-buffer (10×), 3 μL MgSO4 (25 mM), 5 μL dNTP mix (2 mM each)), 1 μL (10 μM) of reverse primer, and 33.8 μL sterile water (total volume 50 μL). PCR conditions were: 95 °C for 2 min followed by 25 cycles of 95 °C for 20 sec, 55 °C for 10 sec, and 70 °C for 15 sec. We then purified the approximately 700 bp PCR amplicons using the MSB Spin PCRapace kit (Invitek, Berlin, Germany).

In the second step, universal primers were used with the following sequences: forward primer, *‘5-TCGTCGGCAGCGTCAGATGTGTATAAGAGACAGCCTACGGGAGGCAGCAG’* (broadly conserved bacterial primer 357F) and reverse primer, *‘5- GTCTCGTGGGCTCGGAGATGTGTATAAGAGACAGTACNVGGGTATCTAAKCC’* (broadly conserved bacterial primer 802R (with adaptations), appended with Illumina adaptor sequences. The PCR amplification mixture contained: 5 μL purified DNA from the first PCR step, 0.1 μL forward primer (10 μM), 14 μL master mix (1 μL KOD Hot Start DNA Polymerase (1 U/μL; Novagen, Madison, WI, USA), 5 μL KOD-buffer (10×), 3 μL MgSO4 (25 mM), 5 μL dNTP mix (2 mM each)), 1 μL (10 μM) of reverse primer, and 30.8 μL sterile water (total volume 50 μL). PCR conditions were: 95 °C for 2 min followed by 25 cycles of 95 °C for 20 sec, 55 °C for 10 sec, and 70 °C for 15 sec. The PCR amplicons were then purified using the MSB Spin PCRapace kit (Invitek, Berlin, Germany).

#### 2.6.3. Library Preparation and 16S MiSeq Sequencing

For the library PCR step in combination with sample-specific barcoded primers, purified PCR products were shipped to BaseClear BV (Leiden, The Netherlands). PCR products were purified, checked on a Bioanalyzer (Agilent), and quantified. This was followed by multiplexing, clustering, and sequencing on an Illumina MiSeq with the paired-end (2×) 300 bp protocol and indexing. The sequencing run was analysed with the Illumina CASAVA pipeline (v1.8.3) by demultiplexing based on sample-specific barcodes. From the raw sequencing data, low quality of sequence reads, reads containing adaptor sequences, or PhiX control with an in-house filtering protocol were discarded and only “passing filter” reads were selected. On the remaining reads, we performed a quality assessment using the FASTQC quality control tool version 0.10.0. (http://www.bioinformatics.babraham.ac.uk/projects/fastqc/, accessed on 8 February 2022).

Sequences of the 16S rRNA gene were analysed using a workflow based on QIIME 1.8 [20]. On average, 23,918 (range 11,071–36,288) 16S rRNA gene sequences per sample were analysed. We performed operational taxonomic unit (OTU) clustering (open reference), taxonomic assignment and reference alignment with the pick_open_reference_otus.py workflow script of QIIME, using uclust as clustering method (97% identity) and GreenGenes v13.8 [21,22,23] as reference database for taxonomic assignment. Reference-based chimera removal was performed with Uchime [24]. The RDP classifier version 2.2 was performed for taxonomic classification [25].

### 2.7. Calculation and Statistical Analyses

Weight, length, and BMI values were converted to sex- and age-adjusted standard deviation scores (SDS) using the UK 1990 growth reference at birth and WHO growth standards at later time points (LMS Growth [26]). Internal SDS were calculated for each skinfold thickness site by calculating the residuals from linear regression models, adjusted for sex and age, and then the mean skinfolds SDS across the four sites was calculated as a measure of infant adiposity.

Low- vs high-butyrate groups were arbitrarily defined based on the median of HM butyrate concentrations.

Continuous variables were summarised as mean ± standard deviation or median (interquartile range) and categorical variables as number (%).

Multiple linear regression models were run with HM butyrate concentration at 6 weeks (when all infants were still exclusively breastfed) as the predictor and infant growth gains (expressed as SDS changes) as outcomes, including infant sex, birth weight SDS, GA, and postnatal age at visit (in days) as covariates.

To capitalise on the longitudinal growth and macronutrient intake data with appropriate handling of missing values, linear mixed-effects models were used to examine the associations between butyrate concentration with anthropometry and body composition parameters, i.e., weight, height, BMI, and mean skinfolds. The models were adjusted for the same covariates as above, additionally with 0–3 months feeding history (exclusively breastfed vs mixed-fed) with further correction for HM intake volume in sensitivity analysis.

For HM microbiota analyses, multivariate redundancy analyses (RDAs) were performed on 69 samples by 16S rRNA gene sequencing in Canoco version 5.11 using default settings of the analysis type “Constrained” [27]. Relative abundance values of genera were used as response data, and metadata as explanatory variable. Variation explained by the explanatory variables corresponds to the classical coefficient of determination (R2) and was adjusted for degrees of freedom (for explanatory variables) and the number of cases. Canoco determined RDA significance by permutating (Monte Carlo) the sample status. To assess microbiota composition differences between samples with relatively high vs. low butyrate levels, samples were split in two equal groups based on BM butyrate concentration (with ranges as follow: 0.17–0.75 mg/dL and 0.8–2.65 mg/dL for low- vs high-butyrate, respectively). A nonparametric Mann–Whitney U test (two-tailed) was applied on all taxa, as implemented in Graphpad Prism 5.01 (San Diego, CA, USA). FDR correction for multiple testing was applied, unless stated otherwise.

All analyses were performed using SPSS version 25 (IBM Corp, Armonk, NY, USA) and R version 3.6.1 (R Foundation for Statistical Computing, Vienna, Austria). In all analyses, *p* values < 0.1 were considered statistical trends; *p* values < 0.05 indicated statistical significance.

## 3. Results

In total, 71 singleton and full-term born healthy infants were included in the longitudinal models analysing associations between HM butyrate and growth. Of these, 47 had complete measurements of HM intake volume and HM microbiota at 6 weeks of age. All 71 infant participants provided stool samples; two were excluded from microbiome analysis due to having too low read count after sequencing. Of the 69 infant participants included in the microbiome analysis, 56 had butyrate measurements over time (Appendix A). Table 1 presents the baseline characteristics of the population analysed, participants with butyrate concentrations at 6 weeks of age, and participants with butyrate intake measurements between 4–6 weeks of age.

### 3.1. Associations between Maternal/Infant Factors and HM Butyrate Concentration

Table 2 shows HM butyrate concentrations over time. No statistically significant associations between HM butyrate concentration at 6 weeks of age and any baseline maternal, antenatal, or any infant parameter (including maternal age, prepregnancy BMI, height, parity, ethnicity, gestational age, infant sex, and exclusive breastfeeding (EBF) duration) were detected (Appendix A, Appendix A). Among mothers who continued EBF for at least 3 months, HM butyrate concentrations increased with infant age, from 0.76 mg/100 mL at 2 weeks to 1.42 mg/100 mL at 3 months of age (*p* = 0.027, Table 2).

### 3.2. Characterisation of HM Microbiota and Associations with HM Butyrate Concentrations

Overall, microbiota composition in HM samples collected at 6 weeks comprised typical taxa previously reported in HM, with characteristic high relative abundance of various skin-associated bacteria, such as *Staphylococcus*, *Streptococcus*, and *Cutibacterium,* as well as *Acinetobacter* and *Lactococcus* [28,29] (Appendix A).

RDA on the genus level showed that HM butyrate concentration was associated with HM microbiota composition (variation explained 1.07%; *p* = 0.036) (Figure 1a). From the taxa indicated by RDA to be associated with HM butyrate, HM *Acinetobacter* relative abundance indeed showed a positive trend with HM butyrate concentration. The relative abundance of *Acinetobacter* was slightly higher in HM samples with relatively high butyrate levels compared to HM samples with relatively low butyrate levels (non-adjusted *p* = 0.086, Figure 1b). However, no statistically significant association between HM butyrate and any taxon was detected, including typical butyrate-producing bacteria (data not shown).

### 3.3. Associations between HM Butyrate, HM Intake Volume, and Infant Growth

An overall negative relationship between HM butyrate concentration and infant weight (−0.60 + 0.23 SDS/g/100 mL, *p* = 0.01) was detected using longitudinal modelling of all repeated HM butyrate concentrations and infant growth from birth to age 12 months (Table 3, Appendix A). The relationship weakened with increasing age (*p-interaction* = 0.02). A similar relationship between HM butyrate and infant BMI was observed (Table 3).

We explored further relationships with HM butyrate concentrations and intakes at age 6 weeks. At 6 weeks, HM butyrate concentration correlated negatively with HM intake volume (Pearson R = −0.29, *p* = 0.047, Figure 2). After 6 weeks, 14 infants discontinued EBF at 6–12 weeks, 52 discontinued EBF at 3–6 months, and 28 continued EBF for at least 6 months. At 6 weeks, butyrate concentrations (median [IQR]) were comparable between those who discontinued or continued EBF from 6–12 weeks (0.84 [0.91] vs. 0.8 [0.67], *p* = 0.3 Wilcoxon test).

Moreover, cross-sectional HM butyrate concentrations at 6 weeks, but not intakes, were inversely associated with weight and height gains from 0–6 weeks (Table 4, Figure 3). When corrected for HM intake volume, the significant associations between HM butyrate levels and early growth gains were no longer detected (data not shown). HM butyrate concentrations and HM butyrate intakes at age 6 weeks were positively associated with growth and adiposity from 6 weeks to 12 months (Table 4), consistent with a wea-kening negative relationship between HM butyrate and growth with age (Table 3).

## 4. Discussion

Butyrate is one of the SCFA detected in the gut as a product of bacterial fermentation of undigested dietary fibres [30]. However, the origin and role of butyrate in HM is not yet well understood. This study explored the potential origin of HM butyrate from HM microbiota. Previous studies have demonstrated local/intestinal butyrate is likely produced by Clostridiales-dominant microbiota such as *Faecalibacterium prausnitzii*, *Eubacterium rectale*, or *Roseburia intestinalis* [31,32]. However, butyrate in HM may likely be produced through a different route, since the RDA in this study found no significant associations between HM butyrate concentration and the relative abundance of previously reported butyrate producers. A trend of association between the relative abundance of *Oscillospira* (known as a butyrate producer [33,34]) and HM butyrate was observed; however, the other typical butyrate producer, *Faecalibacterium* [35], displayed null association with HM SCFA and was not detected among the top 20 taxa (Figure 1). As butyrate-producing bacteria in the gut microbial community typically are anaerobes, their presence in HM might not be readily expected. In addition, the microbiota profiling method used in this study only assessed the relative abundance of bacterial groups and did not reflect bacterial metabolic activity represented through bacterial gene expression. Therefore, increased bacterial metabolism might have contributed to the increased butyrate levels instead of changes in bacterial community composition. Alternatively, butyrate might have passed from the maternal circulation into HM, but this was beyond the scope of the current study.

There was a positive trend between butyrate concentrations and non-butyrate-producing bacterial taxa in HM, such as *Acinetobacter*. A recently published study detected an acetyl-CoA (ACoA) pathway, known as the main butyrate-producing pathway, in *Acinetobacter* strains [36]. However, to the best of our knowledge, there is no evidence of actual production of butyrate by *Acinetobacter*, and therefore this genus is not categorized yet as a typical butyrate producer.

*Acinetobacter* has also been consistently identified in HM microbiota [37], especially in samples that were collected without preceding aseptic cleansing to the breast [38]. In this study, HM sampling for microbiota analysis was performed using a careful aseptic technique, under direct supervision of the paediatric research nurses.

Abundance of *Acinetobacter* in HM microbiota has been associated with food allergy in infants [39]. This might be related to its influence (as a member of the HM microbiota) on infant gut microbiota development or its interactions with other bacterial groups in the infant intestinal tract [40]. However, *Acinetobacter* specific effects on overall infant health are not well studied yet. In addition, distinct strains of *Acinetobacter* have been shown to be susceptible to direct antimicrobial effects of butyrate [41], and it might therefore be surprising that increased butyrate levels in HM were associated with higher abundance of *Acinetobacter*. However, the balance of antimicrobial effects of HM compounds such as butyrate but also HMOs and Lactoferrin [42,43] affecting *Acinetobacter* as well as other bacterial species might have resulted in changes in overall HM microbiota composition indirectly leading to a net increase in *Acinetobacter* abundance.

Another aim of this study was to examine if butyrate in HM influences HM intake and ultimately infant weight and adiposity. In the current analysis, longitudinal models displayed overall negative associations between HM butyrate concentrations and measures of infant weight and adiposity, similar to our previous report [1], which could potentially prevent excess weight gain and obesity during childhood. From both animal and human studies, butyrate and its producing bacteria have been linked to a lower risk of obesity and metabolic complications, including liver fibrosis [6] and insulin resistance [44,45]. Butyrate may also act as an anti-inflammatory mediator in metabolic diseases [4]. In a piglet model, butyrate appeared to influence lipid metabolism by accentuating adipogenesis and lipid accumulation, possibly via glucose uptake upregulation and de novo lipid production [46]. Furthermore, serving as the source of energy for colonocytes, butyrate may affect energy intake and energy balance, as 10% of energy intake may be attributed to dietary residues entering the large intestine [47]. Other additional unknown mechanisms might also underlie the inverse relationship between HM butyrate and infant growth.

Cross-sectionally, when examining butyrate intakes through HM rather than concentrations, the inverse associations between butyrate and early growth became less visible. Since HM butyrate concentration was inversely correlated with HM intake volume, it could be speculated that the associations between butyrate content and early growth were either mediated or confounded by lower HM intake, i.e., the high butyrate concentration in HM might be the reason for low HM intake in some infants. Some recent animal studies [48,49] have reported that acute oral butyrate administration via intragastric gavage rapidly induced satiety and decreased food intake in mice, presumably via modulation of neuropeptide XY neurons via vagal nerves [50]. In addition, other SCFAs such as propionate have also been reported to be key molecules governing the signaling pathway within the gut–brain axis and influencing appetite [51]. Consequently, the interplay between butyrate odor and/or taste in HM and its effect on appetite regulation may potentially lower infant HM intake and contribute to attenuated HM intake and early infant weight gain.

In this study, negative associations between HM butyrate and infant weight and BMI seemed to be stronger at early, rather than later, infant ages. Since infants in this study were solely dependent on HM consumption during the first 6 weeks of age, the mechanistic pathway of butyrate on early growth modulation could be mainly speculated to occur via appetite regulation and HM intake reduction.

Moreover, in our longitudinal models (Table 3), HM butyrate was overall negatively associated with growth, but with a positive interaction with age (butyrate*age), indicating a weakening in this negative relationship, and it is likely that positive “catch-up” growth occurs as infants are introduced to other forms of nutrition.

Reflecting on the current setup, the strengths of this study include the estimation of butyrate intake alongside its concentration measurement by quantifying HM intake volume, which is not routinely included in many HM research cohorts. To the best of our knowledge, this is the only study that investigates the links between HM butyrate concentration, HM intake volume, and HM microbiota. The longitudinal design of this study also enabled us to analyse the associations between HM butyrate and subsequent weight and adiposity gains during infancy.

However, although applying strict inclusion criteria allowed us to omit some confounding factors, e.g., mode of delivery and antibiotics use during antenatal period, the relatively small numbers of samples have limited sensitivity analyses in this study. Although a lot of antenatal/maternal information was recorded through the perinatal questionnaire, detailed maternal diet during the breastfeeding period that might influence butyrate levels in HM was not available. Future large longitudinal studies with more complete maternal data are needed to examine the link between HM butyrate, HM intake volume, and growth gains during infancy in more detail.

## 5. Conclusions

In this current infant cohort, we observed a weak association between HM butyrate and HM microbiota composition. The lack of relationship between HM butyrate and its typical bacterial taxa producers might suggest alternative sources of butyrate in HM, such as maternal transfer. However, changes in butyrate concentration in HM might in turn have modulated HM microbiota composition through antimicrobial effects. We also observed an overall negative influence of HM butyrate on early infant weight and adiposity gains, which might have potentially been mediated by appetite modulation and decreased HM intake volumes.

## Figures and Tables

**Figure 1 nutrients-15-00916-f001:**
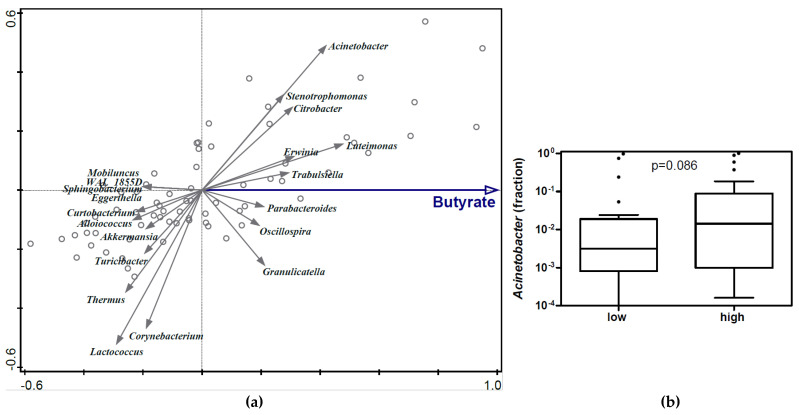
Associations between HM butyrate concentration and HM microbiota. (**a**) Redundancy analysis (RDA) on the genus level, assessing the associations between HM butyrate concentrations and HM microbiota composition. Genera were used as response data and butyrate concentration as explanatory data. Variation of HM microbiota composition explained by butyrate was 1.07% (*p* = 0.036); (**b**) Relative abundance of *Acinetobacter* in HM microbiota composition based on HM butyrate concentrations. Mann–Whitney test was used to compare the relative abundance of *Acinetobacter* in HM between high- vs. low-butyrate groups (arbitrarily defined). Boxplots are displayed as Tukey whiskers.

**Figure 2 nutrients-15-00916-f002:**
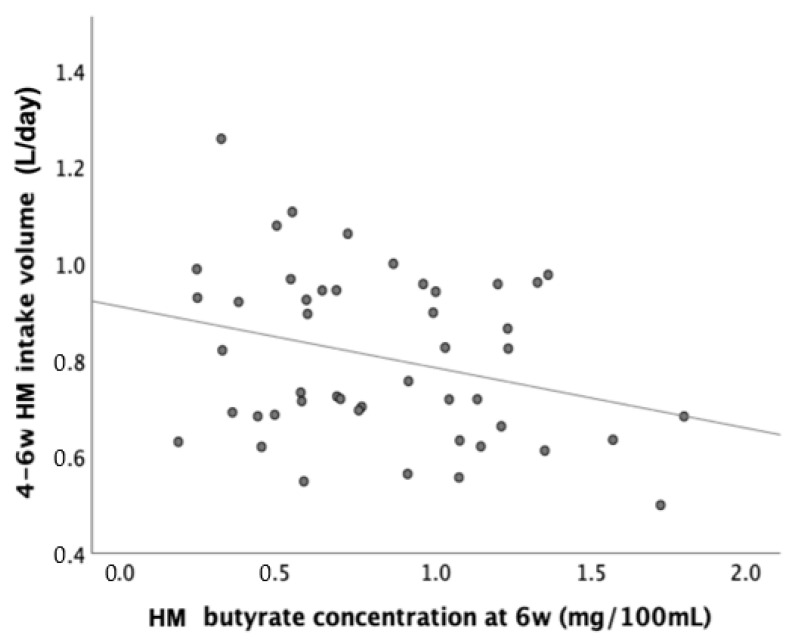
Inverse association between HM butyrate concentration and HM intake volume at 4–6 weeks (*p* = 0.047). HM = human milk, w = weeks.

**Figure 3 nutrients-15-00916-f003:**
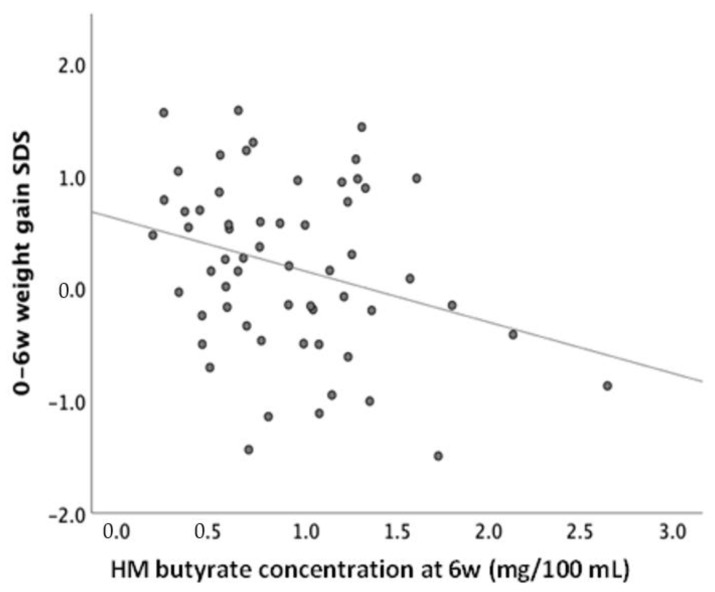
Negative association between HM butyrate level and early infant weight gain (0−6 weeks) (*p* = 0.04). HM = human milk, SDS = standard deviation scores, w = weeks.

**Table 1 nutrients-15-00916-t001:** Baseline characteristics of mother–infant dyads based on sample availability.

	All Subjects Included in Longitudinal Analyses between HM Butyrate and Growth (Total *n* = 71)	Subjects with HM Butyrate Concentration Measured at 6 Weeks (Total *n* = 59)	Subjects with HM Butyrate Intake Measured between 4–6 Weeks (Total *n* = 47)
Gestational age (weeks)	40.3 ± 1.1	40.3 ± 1.1	40.4 ± 1.2
Maternal age (years)	33.2 ± 4.5	33.0 ± 4.7	33.3 ± 4.4
Maternal prepregnancy BMI (kg/m^2^)	22.4 ± 2.5	22.3 ± 2.6	22.5 ± 2.8
Maternal height (cm)	166.8 ± 6.9	167.2 ± 6.9	167.4 ± 7.1
Maternal parity (% primiparous)	41%	41%	38%
Maternal ethnicity (% European)	92%	92%	94%
Infant sex (% male)	61%	63%	60%
Infant birth weight SDS	0.15 ± 0.76	0.16 ± 0.74	0.23 ± 0.78
Infant birth length SDS	−0.17 ± 0.74	−0.17 ± 0.73	−0.17 ± 0.79

Values are mean ± SD. SDS values are based on UK 1990 growth reference.

**Table 2 nutrients-15-00916-t002:** HM butyrate concentrations over time.

	Age			
	2 Weeks	6 Weeks	3 Months	6 Months
	*n* = 31	*n* = 49	*n* = 32	*n* = 25
Butyrate (mg/100 mL)	0.76 ± 0.45	0.85 ± 0.41	1.42 ± 0.91	1.27 ± 0.94

Values are mean ± SD. Only includes HM samples of mothers who exclusively breastfed their infants for at least 3 months.

**Table 3 nutrients-15-00916-t003:** Longitudinal associations between repeated measures of HM butyrate concentrations and infant growth parameters.

Growth Parameters	Butyrate		Butyrate*Age	
	B ± SE	*p*	B ± SE	*p*
Weight SDS	−0.58 ± 0.22	**0.01**	0.12 ± 0.05	**0.02**
Length SDS	−0.22 ± 23	0.34	0.04 ± 0.05	0.49
BMI SDS	−0.66 ± 0.29	**0.02**	0.14 ± 0.07	**0.03**
Mean SF SDS	−0.06 ± 0.3	0.84	−0.01 ± 0.07	0.92

HM butyrate concentrations and infant growth were measured at 6 time points, i.e., birth, 2 weeks, 6 weeks, 3 months, 6 months, and 12 months. All models were adjusted for infant sex, gestational age, visit time points, birth weight SDS, and 0–3 months feeding history (exclusively breastfed vs. mixed-fed). Significant *p* values are highlighted in bold.

**Table 4 nutrients-15-00916-t004:** Associations between HM butyrate concentrations/intakes and infant growth parameters.

Growth Parameters	Predictors			
	Butyrate Concentration at 6 Weeks		Butyrate Intake at 4–6 Weeks	
	B ± SE	*p*	B ± SE	*p*
**0–6 weeks**				
Weight gain SDS	−0.40 ± 0.19	**0.04**	−0.02 ± 0.04	0.67
Length gain SDS	−0.39 ± 0.18	**0.04**	−0.03 ± 0.04	0.34
BMI gain SDS	−0.31 ± 0.27	0.25	−0.003 ± 0.05	0.94
Mean SF gain SDS	−0.34 ± 0.27	0.22	0.004 ± 0.05	0.95
**6 weeks–12 months**				
Weight gain SDS	0.47 ± 0.21	**0.03**	0.07 ± 0.04	0.055
Length gain SDS	0.04 ± 0.21	0.85	−0.03 ± 0.04	0.47
BMI gain SDS	0.5 ± 0.22	**0.03**	0.11 ± 0.04	**0.005**
Mean SF gain SDS	0.49 ± 0.25	**0.05**	0.004 ± 0.05	**0.02**

Unstandardised estimates ± standard errors are displayed. All models are adjusted for infant sex, gestational age, postnatal age at visit, and birth weight SDS. “6 weeks–12 months” models are additionally adjusted for 0–3 months feeding history (exclusively breastfed vs. mixed-fed). Significant *p* values are highlighted in bold.

## Data Availability

The data that support the findings of this study are available on request from Professor Ken Ong (ken.ong@mrc-epid.cam.ac.uk).

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
