# Peer review of "Butyrate in Human Milk: Associations with Milk Microbiota, Milk Intake Volume, and Infant Growth"

_nutrients, 2023, doi:10.3390/nu15040916_

Round 1

Reviewer 1 Report

In this manuscript, the hypothesis of butyrate in human milk coming from its microbiota has been rejected and some non-expected associations between butyrate concentrations and bacterial taxa have been found. However, the authors consider in the text alternative sources of this compound in human milk and possible reasons for the modulation of its concentration. Because butyrate has been shown to protect against obesity and insulin resistance through its anti-inflammatory properties, would it be worth determining any inflammation biomarkers in the babies to investigate the underlying mechanism?

Figure 1b) Relative abundance of Acinetobacter in HM microbiota composition based on HM butyrate concentrations. The authors should explain the information extracted from this figure.

Minor comment:

In paragraph 3.2 the first sentence refers to Supplementary Table 1 but it should refer to Supplementary Figure 1. 

Author Response

We would like to thank reviewer 1 for all their helpful comments which we believe have helped us to improve the manuscript.

In this manuscript, the hypothesis of butyrate in human milk coming from its microbiota has been rejected and some non-expected associations between butyrate concentrations and bacterial taxa have been found. However, the authors consider in the text alternative sources of this compound in human milk and possible reasons for the modulation of its concentration. Because butyrate has been shown to protect against obesity and insulin resistance through its anti-inflammatory properties, would it be worth determining any inflammation biomarkers in the babies to investigate the underlying mechanism?

Thanks for this comment. We would like to reiterate that we do not fully reject the hypothesis that “butyrate in human milk (HM) coming from HM microbiota” since we observed the association between HM butyrate and HM microbiota composition at 6 weeks (p=0.036) although no association with the abundance of typical butyrate producers was detected. Although the reviewer’s suggestion to determine any infant inflammation biomarkers in this study in order to shed more light on the underlying mechanism, we are afraid that this might be too speculative, not feasible (due to sample size), and would be beyond the scope of this study.

Figure 1b) Relative abundance of Acinetobacter in HM microbiota composition based on HM butyrate concentrations. The authors should explain the information extracted from this figure.

We have mentioned the information extracted from this figure in Results section (line 272-274) and have now provided more explanation in Discussion (line 350-370).

Minor comment:

In paragraph 3.2 the first sentence refers to Supplementary Table 1 but it should refer to Supplementary Figure 1. 

Thanks for pointing this out - this has now been corrected.

Reviewer 2 Report

This paper explores how HM butyrate is associated with milk microbiota, milk intake and infant growth. The paper provides some interesting results supporting, that HM butyrate has a role in development of infant adiposity.

 A limitation, as also mentioned by the authors, is the small sample size, and also that the number of infants in the different analysis differs very much, as there were quite some infants who missed several of the measurements. In Supplementary table 2 it is outlined that there are 5 different numbers included in different analysis from 71 to 47. I was surprised that in the abstract only n=59 was mentioned. In table 2 it is mentioned that there were only 31 buturate measurements at 2 weeks and only 25 at 6 months. Why are 12 mo measurements not mentioned in this table?

In table 3 there is a column with butyrate and one with butyrate*age, but I miss in the table text  a clear explanation of these two columns.  

L 313-316. It would be good to have a better explanation of why a positive association with growth and adiposity is consistent with a weakening negative relationship.

In Fig 2 and 3 it would be helpful to have the p-value for the associations in the figure text.

Supplementary Figure 2 hase no Fig text and there is no mention of this figure in the text.  

Author Response

We would like to thank reviewer 2 for all their helpful comments which we believe have helped us to improve the manuscript.

This paper explores how HM butyrate is associated with milk microbiota, milk intake and infant growth. The paper provides some interesting results supporting, that HM butyrate has a role in development of infant adiposity.

Many thanks for this assessment of our study.

A limitation, as also mentioned by the authors, is the small sample size, and also that the number of infants in the different analysis differs very much, as there were quite some infants who missed several of the measurements. In Supplementary table 2 it is outlined that there are 5 different numbers included in different analysis from 71 to 47. I was surprised that in the abstract only n=59 was mentioned. In table 2 it is mentioned that there were only 31 buturate measurements at 2 weeks and only 25 at 6 months. Why are 12 mo measurements not mentioned in this table?

We apologise to have confused the reviewer regarding our sample size. We found conducting such study (longitudinal infant cohort with extensive breast milk investigation) challenging, especially with regard to obtain complete samples from all participants. N=59 in the abstract refers to number of subjects who received at least 6 week-period of exclusive breastfeeding duration.

Regarding Table 2, birth and 12 mo butyrate measurements were not included due to too little sample sizes (this is expected since rate of breasteeding in the UK beyond 6 mo of age is too low) that might lead to unnecessary confusion to the readers. We believe this should be appropriate to not include both butyrate levels measured in HM at birth at 12 mo since we only used longitudinal butyrate measurements in linear mixed-effect models (Table 3) when associating with longitudinal growth trajectories (chosen due to appropriate handling of missing values) while the other analyses only used butyrate levels at either 6 weeks (cross-sectionally with HM microbiota - Figure 1; HM intake volume - Figure 2; and snapshots of 2 growth periods - Table 4) or 2 weeks-6 months when associating with contemporaneous growth parameters (Supplementary Table 4).

In table 3 there is a column with butyrate and one with butyrate*age, but I miss in the table text  a clear explanation of these two columns.  

The explanation is available in Discussion (Line 404-407) and we have now made this more explicit.

L 313-316. It would be good to have a better explanation of why a positive association with growth and adiposity is consistent with a weakening negative relationship.

We have mentioned this in the previous sub-section of the result (Table 3, Line 289-293) and we have now made this more explicit.

In Fig 2 and 3 it would be helpful to have the p-value for the associations in the figure text.

We have now added p values to both figures.

Supplementary Figure 2 has no Fig text and there is no mention of this figure in the text.  

We have now mentioned Supplementary Figure 2 in the text (Line 257).